# Peer review of "Position-Dependent Mass Quantum systems and ADM formalism"

_SciPost Physics Proceedings_

## Round 2 · Referee Report · Anonymous · 2020-11-1

Report

In the revised version the author has fulfilled all my raised points, and thus I recommend the paper for publication. Some expressions should be replaced by more formal forms (e.g. wanna→want, etc) and moreover English editing is needed at the proof stage.

---

## Round 2 · Author Response

To
The Editor
SciPost Physics Proceedings
Dear Editor,

We thank the editor and the referee for giving us the opportunities once to address some questions/comments and helping us to improve our presentation. We have modified the paper to accommodate the referee's suggestions accordingly. We have also corrected possible typos/errors.

We hope that the editor does appreciate our efforts to make sure that the paper is of highest quality. We are confident that we have now answered, in detail, to all the points raised by the referee and henceforth resubmit the paper for publication.
Our replies to the referee's comments are given below.
Best regards,
Authors
[Comment-\#1]{The manuscript “Position-Dependent Mass Quantum systems and ADM formalism”, by Davood Momeni
tries to present a Hamiltonian formulation of general relativity, inspired by position-dependent mass systems, and then it proposes a quantum version of the theory.\\
The Hamiltonian formulation of general relativity has been discussed extensively in the literature for decades. Some references are given, but many important contributions are missing (one can find them in textbooks or Review papers). In any case, the author should tone down his claims about his findings and their significance, and remove statements such as “this is the first time in literature when a first order Hamiltonian version of the gravitational field equations.”}
[Answer-\#1]{We thank the referee for pointing out this remark. In this revision, we have carefully corrected this point as suggested . Please see the discussion after eq. (3.10).}

[Comment-\#2]{The author’s approach is based on the “super mass tensor”. However, the author defines this quantity as a derivative on the Lagrangian, and hence expression (2.5) is a loop definition. }
[Answer-\#2]{We thank the referee for pointing out this remark. In this revision, we have carefully corrected this point as suggested . Please see the discussion after eqs. (2.5), (2.6).}

[Comment-\#3]{I cannot see how the complicated form of field equations (3.10) can be helpful.}
[Answer-\#3]{The mentioned Hamilton equations is considered as a covariant generalization of the momentum evolutionary equation in the standard ADM decomposition. The only difference is here we don't specify special time foliation of the spacetime, i.e, no need to consider $x^{0}=constant$ hypersurfaces as it is very convenient in ADM method. A part of this foliastion freedom, as we showed that the equation reduces to the momentum time equation if one adapts a preferred time foliation.}

[Comment-\#4]{The matter sector, which is crucial in GR since it is the source of non-trivial curvature and geometry, is missing from the discussion.}
[Answer-\#4]{We thank the referee for pointing out this remark. In this revision, we have carefully corrected this point as suggested . Please see the discussion after eq. (1.1).}

[Comment-\#5]{The author faces the problem as a simple quantum mechanical problem and not as a quantum field theoretical one, and thus the discussion on renormalizability etc is missing.}
[Answer-\#5]{We should emphasis here that theory which we studied in this paper is considered as an attempt to construct quantum mechanics on a classical GR background. There is no simple field theoretic interpretation for the Hamiltonian which we obtained in this work as well as any other brackets are simply non quantum field theoretical one. In his approach, we can't reach the renormalizability as it has been investigated in many other alternative quantum gravity scenarios.}

[Comment-\#6]{Solutions (5.7),(5.8) do not have an obvious meaning, and in any case it is strange that the author finds non-trivial structure in the absence of matter.}
[Answer-\#6]{We added a discussion about these equations after eqs.(5.8).}

[Comment-\#7]{The English of the manuscript need editing.\\
In summary, a radical modification is needed before I will be able to reconsider the manuscript for publication.}
[Answer-\#7]{We fixed several typos and improved the presentation of the paper..}

---

## Round 2 · List of Changes

[Comment-\#1]{The manuscript “Position-Dependent Mass Quantum systems and ADM formalism”, by Davood Momeni
tries to present a Hamiltonian formulation of general relativity, inspired by position-dependent mass systems, and then it proposes a quantum version of the theory.\\
The Hamiltonian formulation of general relativity has been discussed extensively in the literature for decades. Some references are given, but many important contributions are missing (one can find them in textbooks or Review papers). In any case, the author should tone down his claims about his findings and their significance, and remove statements such as “this is the first time in literature when a first order Hamiltonian version of the gravitational field equations.”}
[Answer-\#1]{We thank the referee for pointing out this remark. In this revision, we have carefully corrected this point as suggested . Please see the discussion after eq. (3.10).}

[Comment-\#2]{The author’s approach is based on the “super mass tensor”. However, the author defines this quantity as a derivative on the Lagrangian, and hence expression (2.5) is a loop definition. }
[Answer-\#2]{We thank the referee for pointing out this remark. In this revision, we have carefully corrected this point as suggested . Please see the discussion after eqs. (2.5), (2.6).}

[Comment-\#3]{I cannot see how the complicated form of field equations (3.10) can be helpful.}
[Answer-\#3]{The mentioned Hamilton equations is considered as a covariant generalization of the momentum evolutionary equation in the standard ADM decomposition. The only difference is here we don't specify special time foliation of the spacetime, i.e, no need to consider $x^{0}=constant$ hypersurfaces as it is very convenient in ADM method. A part of this foliastion freedom, as we showed that the equation reduces to the momentum time equation if one adapts a preferred time foliation.}

[Comment-\#4]{The matter sector, which is crucial in GR since it is the source of non-trivial curvature and geometry, is missing from the discussion.}
[Answer-\#4]{We thank the referee for pointing out this remark. In this revision, we have carefully corrected this point as suggested . Please see the discussion after eq. (1.1).}

[Comment-\#5]{The author faces the problem as a simple quantum mechanical problem and not as a quantum field theoretical one, and thus the discussion on renormalizability etc is missing.}
[Answer-\#5]{We should emphasis here that theory which we studied in this paper is considered as an attempt to construct quantum mechanics on a classical GR background. There is no simple field theoretic interpretation for the Hamiltonian which we obtained in this work as well as any other brackets are simply non quantum field theoretical one. In his approach, we can't reach the renormalizability as it has been investigated in many other alternative quantum gravity scenarios.}

[Comment-\#6]{Solutions (5.7),(5.8) do not have an obvious meaning, and in any case it is strange that the author finds non-trivial structure in the absence of matter.}
[Answer-\#6]{We added a discussion about these equations after eqs.(5.8).}

[Comment-\#7]{The English of the manuscript need editing.\\
In summary, a radical modification is needed before I will be able to reconsider the manuscript for publication.}
[Answer-\#7]{We fixed several typos and improved the presentation of the paper..}

You are currently on this page

Resubmission 2008.02113v2 on 29 October 2020

---

## Editorial Decision

publication_decision_taken:_accept